# Transcriptome Analysis Provides Insights into Korean Pine Tree Aging and Response to Shading

**Zha-Long Ye [1], Jin-Yi Liu [2], Jian Feng [3,\*] and Wan-Feng Li [1,\*]**

1   State Key Laboratory of Tree Genetics and Breeding, Key Laboratory of Tree Breeding and Cultivation of the National Forestry and Grassland Administration, Research Institute of Forestry, Chinese Academy of Forestry, Beijing 100091, China; kemiye@caf.ac.cn
2   State-Owned Dabiangou Forest Farm in Qingyuan Man Autonomous County, Fushun 113000, China; ljy3052909@163.com
3   Liaoning Academy of Forestry Science, Shenyang 110000, China
*   Correspondence: fengjian-0205@163.com (J.F.); liwf@caf.ac.cn (W.-F.L.); Tel.: +86-10-6288-9628 (W.-F.L.); Fax: +86-10-6287-2015 (W.-F.L.)

**Abstract:** Age controls a tree's responses to environmental cues and shading influences tree growth and physiology. These are basic principles of "Afforestation under canopy", an approach that is widely used in the regeneration of Korean pine forests. Studying the underlying mechanism helps us to understand tree adaptation and utilize it in forest management. In this study, we investigated the transcriptomic changes in the uppermost main stems of the Korean pine tree (*Pinus koraiensis*, Sieb. et Zucc.) at different ages (5, 7, 10, 14, and 17 years) and in different growth conditions (open-grown and shade-grown trees) using RNA-Seq. In total, 434,005,837 reads were produced and assembled into 111,786 unigenes. After pairwise comparisons, 568 differentially expressed unigenes (DEUs) were identified. The greatest number of DEUs was identified in the comparison between 5-year-old open-grown trees and 17-year-old shade-grown trees, while no DEUs were identified in 15 pairwise comparisons. Among these 568 DEUs, 45 were assigned to gene ontology (GO) terms associated with response to environmental changes, including "response to stress" (26) and "response to light and temperature" (19); 12 were assigned to GO terms associated with sexual reproduction, such as "sexual reproduction", "specification of floral organ identity", "pollen tube guidance", and "fruit ripening"; 15 were heat shock protein genes and showed decreased expression patterns with age; and one, annotated as *Pinus tabuliformis DEFICIENS-AGAMOUS-LIKE 1*, showed an increased expression pattern with age, independent of the reproductive state or growth conditions of Korean pine trees. Altogether, these findings not only demonstrate the molecular aspects of the developmental and physiological effects of age and shading on Korean pine trees, but also improve our understanding of the basic principles of "Afforestation under canopy".

**Keywords:** adaptation; age; conifer; shading; silviculture; transcriptome

## 1. Introduction

Pollution and climatic degradation have led to our focus on age-related changes in tree development and physiology because trees of different ages respond differently to the environment. For example, after heat stimulation, cambium cells in younger trees resume their activities earlier, showing strong sensitivity to temperature [1]. Moreover, elevated resistance to pathogens occurs with aging in apple trees and other flowering plants [2–5]. Sensitivity to the environmental cues that induce flowering increases with tree age, and this provides a core basis for the separation of the juvenile vegetative phase from the adult vegetative phase [6,7]. These studies indicate that the developmental and physiological responses of trees change with age. However, how age controls the changes in these responses in trees is unknown.

Shading affects light cues, influencing plant growth and development, and this has been well studied in crops. For example, the leaf angle affects a plant's photosynthetic efficiency and yield [8], and the molecular mechanism of this factor has been widely reported in rice and maize, providing a possibility for the quantitative control of leaf angles at different canopy levels [9]. Other methods like density control, dwarfing rootstock [10], and pruning [11], have been studied in fruit tree management and used to manipulate shading to improve yield. Shading also affects plant metabolism. For example, shading decreases the epicatechin, epicatechin gallate, and epigallocatechin levels, as well as increasing the caffeine levels in tea plants [12]; moreover, a higher vitamin C content has been found in blueberry fruit subjected to shading treatment [13].

In forest tree management, shading is also used to control tree growth. Korean pine (*Pinus koraiensis*, Sieb. et Zucc.) is a valuable crop and timber tree, and its pine nuts are a popular snack and raw material for oil production. Therefore, Korean pine is the most popular tree species for farmers in Northeast China; however, it takes about 14 years for a tree grown from a seed to bear fruit, resulting in slow returns. Furthermore, adult Korean pines are photophilous, but juvenile trees are resistant to shade, and proper shading does not suppress the growth of young Korean pines [14]. This habit is used to solve the problem of the slow artificial regeneration of Korean pine forests, known as "afforestation under canopy" (Figure 1). Usually, 4-year-old Korean pine saplings that are more than 35 centimeters tall are selected and planted with two-meter spacing under 30–40-year-old larch trees, which were planted with five-meter spacing.

Shade-grown trees          Open-grown trees

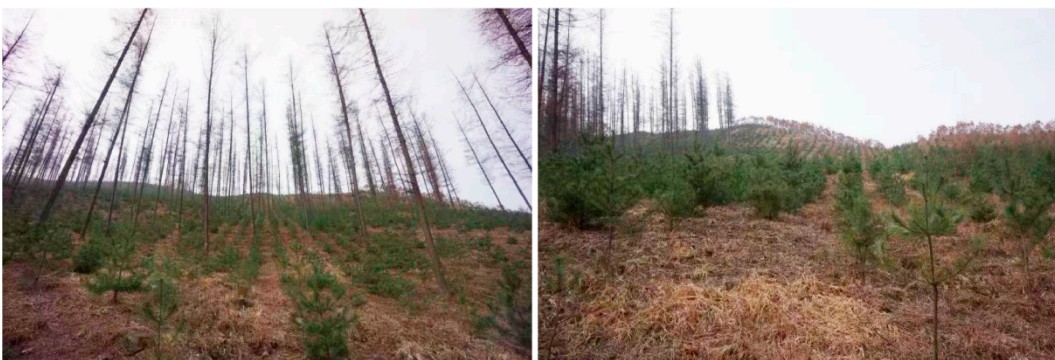

**Figure 1.** Images of shade-grown (**left**) and open-grown (**right**) Korean pine trees. Shading was provided by afforestation under canopy, which is a management strategy for Korean pines used in the Dabiangou seed orchard. Korean pine saplings that are four years old and more than 35 centimeters tall are selected and planted with two-meter spacing under larch trees. The larch trees were 30–40 years old and were planted with five-meter spacing.

We hypothesized that gene expression changes during tree aging, and age and shading effects on the development and physiology of Korean pine trees, could be detected at the transcriptional level. To prove this hypothesis, we investigated the transcriptomes of Korean pines of different ages and the effects of shading on them, providing a theoretical basis for Korean pine management.

## 2. Materials and Methods

### 2.1. Plant Material

Tree development and physiology vary at different heights. The apical stem, to some extent, is considered to represent the physiological age of a tree developed from a seed. To weaken position effects, the uppermost newly produced main stems were collected from 5-, 7-, 10-, 14-, and 17-year-old active Korean pine trees. Active 14- and 17-year-old Korean pine trees that were planted under the canopy were also sampled to study the effect of shading on the development and physiological responses of Korean pine trees. The

trees under study were growing in different stands in the Dabiangou seed orchard (42°02′ N, 125°07′ E), Liaoning Province, in Northeast China (Table 1), and all trees were grown from seeds.

**Table 1.** Information of samples used for RNA sequencing.

| Tree Age | Sample | Replicate | Cone | Tree Height (m) | Current Increment (cm) | Location of the Tree | | |
|---|---|---|---|---|---|---|---|---|
| | | | | | | East Longitude | North Latitude | Altitude (m) |
| 17 years | Pk-17B | 17B-1 | Big | 4.70 | 64 | 125.05 | 42.05 | 538 |
| | | 17B-2 | Small | 4.53 | 69 | 125.05 | 42.05 | 538 |
| | | 17B-3 | Big | 3.60 | 50 | 125.05 | 42.05 | 538 |
| | Pk-17 | 17-1 | - | 4.04 | 66 | 125.05 | 42.05 | 538 |
| | | 17-2 | - | 4.20 | 62 | 125.05 | 42.05 | 538 |
| | | 17-3 | - | 3.24 | 51 | 125.05 | 42.05 | 538 |
| | Pk-17S | 17S-1 | - | 4.40 | 47 | 125.35 | 42.01 | 538 |
| | | 17S-2 | - | 4.63 | 49 | 125.35 | 42.01 | 538 |
| | | 17S-3 | - | 4.52 | 46 | 125.35 | 42.01 | 538 |
| 14 years | Pk-14B | 14B-1 | Small | 3.40 | 72 | 125.07 | 42.02 | 448 |
| | | 14B-2 | Big | 3.04 | 71 | 125.07 | 42.02 | 448 |
| | | 14B-3 | Big | 3.04 | 61 | 125.07 | 42.02 | 448 |
| | Pk-14 | 14-1 | - | 4.00 | 60 | 125.07 | 42.02 | 448 |
| | | 14-2 | - | 2.20 | 54 | 125.07 | 42.02 | 448 |
| | | 14-3 | - | 2.46 | 51 | 125.07 | 42.02 | 448 |
| | Pk-14S | 14S-1 | - | 2.88 | 45 | 125.35 | 42.01 | 538 |
| | | 14S-2 | - | 2.60 | 37 | 125.35 | 42.01 | 538 |
| | | 14S-3 | - | 1.98 | 20 | 125.35 | 42.01 | 538 |
| 10 years | Pk-10 | 10-1 | - | 1.78 | 37 | 125.35 | 42.04 | 525 |
| | | 10-2 | - | 1.77 | 43 | 125.35 | 42.04 | 525 |
| | | 10-3 | - | 1.51 | 41 | 125.35 | 42.04 | 525 |
| 7 years | Pk-7 | 7-1 | - | 0.58 | 12 | 125.27 | 42.08 | 433 |
| | | 7-2 | - | 0.75 | 18 | 125.27 | 42.08 | 433 |
| | | 7-3 | - | 0.63 | 15 | 125.27 | 42.08 | 433 |
| 5 years | Pk-5 | 5-1 | - | 0.70 | 37 | 125.07 | 42.02 | 420 |
| | | 5-2 | - | 0.64 | 26 | 125.07 | 42.02 | 420 |
| | | 5-3 | - | 0.55 | 29 | 125.07 | 42.02 | 420 |

Pk: *Pinus koraiensis* Sieb. et Zucc. Small cones were formed in the current year (2021), and big cones were formed in the previous year (2020). Pk-17S and Pk-14S were shade-grown trees, while others were open-grown trees. Pk-17B and Pk-14B were trees with seed cones, while others were trees without seed cones. - indicates no cone.

Sample collection was performed on the morning of 31 July 2021. Secondary growth mediated by cambium and lignification occurred actively in these sampled stems. When sampling, we found that some 14- and 17-year-old trees bore cones and these were collected separately (Table 1). Each sample had three replicates. After the removal of branches and needles, a 15-cm segment (from the top) of the main stem of each tree was cut, immediately frozen in dry ice, and then stored at −80 °C until RNA extraction.

*2.2. CDNA Library Preparation and Transcriptome Sequencing*

Total RNA extraction from each tree was performed using a plant RNA purification reagent, as per the manufacturer's instructions (Invitrogen, San Diego, CA, USA). RNA was quantified using a Qubit RNA kit (Thermo Fisher Scientific, Inc., Wilmington, DE, USA). Five micrograms of total RNA from each tree was used to generate an RNA library for Illumina paired-end sequencing. In total, 27 libraries were constructed.

After isolation from 5 μg of total RNA, the mRNA was purified using mRNA capture beads and then broken into 300–500 bp short fragments using magnetic beads. Subsequently,

these fragments were employed as templates for the synthesis of first-strand cDNA, using a random hexamer primer. Then, second-strand cDNA was synthesized and cleaned up to prepare for end repair and the addition of poly A. Thereafter, these strands were connected to sequencing adapters and the desired cDNA was isolated using Hieff NGS® DNA Selection Beads (YEASEN, Shanghai, China). To enrich the desired cDNA, PCR amplification was performed. PCR products were tested on 2% agarose gel. Finally, the libraries were sequenced using the Illumina HiSeq™ 2500 platform (Illumina, CA, USA).

### 2.3. Transcriptome Assembly and Annotation

FastQC (http://www.bioinformatics.babraham.ac.uk/projects/fastqc, accessed on 17 November 2022) and Trimmomatic v. 0.32 (https://github.com/timflutre/trimmomatic, accessed on 18 November 2022) were used for quality control. The raw Illumina RNA-Seq reads were pre-processed by discarding reads with adapters, with >5% unknown nucleotides, those of low quality (quality score < 20), or those with <20 bp. De novo assembly was performed using Trinity software v. 2.15 (https://github.com/trinityrnaseq/trinityrnaseq, accessed on 18 November 2022) with the default parameters.

To identify the Korean pine protein-coding genes, blastx was used to search our assembled sequences against the blastx_swissprot, protein, blastp_swissport, Pfam, SignalP, eggnog, blast2GO, Pfam2GO, and Kegg databases with an *E* value $< 1 \times 10^{-5}$. Gene Ontology (GO) terms associated with the top hits in the nine protein databases were assigned to the annotated transcripts. After sequence clustering with Trinity software, unigenes were obtained and used for further analysis.

### 2.4. Identification of Differentially Expressed Unigenes (DEUs)

To identify the DEUs, 27 sets of sequencing reads were mapped pairwise to the assembled reference transcripts, and 36 comparisons were produced. Then, the normalized expression value was calculated as the fragments per kilobase of transcript per million mapped reads (FPKM). A pairwise comparison between two samples was performed to identify the DEUs with a false discovery rate (FDR)-corrected *p* value cutoff of 0.05 and a minimum expression fold change of 2.

### 2.5. Statistical Analyses and Graphics

Multiple comparisons were used to obtain the significance levels through the ANOVA Duncan method using SPSS version 26. The relationships between the gene expression patterns and age were analyzed with simple linear regression using GraphPad Prism version 9. Heatmaps were used to show the gene expression pattern using TBtools v. 2.04 [15], and log and row scales were used to display the changes in patterns between groups.

## 3. Results

### 3.1. Age and Shading Affect Reproductive Phase Change

In nine 14-year-old trees, small or big developing seed cones were observed in three trees, while the other six trees did not bear cones (Table 1). After fertilization, it takes one year for Korean pine trees to form small cones and two years for big cones to form. Therefore, at the age of 12, reproductive phase change had occurred in the trees with big cones. Based on the fruiting state of these nine trees, we concluded that reproductive phase change occurred in a genotype-dependent manner. This phenomenon was also observed in 17-year-old trees (Table 1). However, no cones were observed in 5, 7 or 10-year-old trees (Table 1). Notably, cones were not observed in 14- or 17-year-old trees growing under the canopy either, indicating that the occurrence of reproductive phase change in these trees might be delayed by shading.

### 3.2. Transcriptome Assembly and Annotation

A total of 434,005,837 reads were produced from 27 samples (Table 2), with a 150-bp read length. All clean reads were assembled into 213,276 transcripts, including 111,786

unigenes (Table 3). Then, the 27 sets of reads were aligned to the assembled reference transcripts, and their mapping ratios ranged from 69.67% to 87.89%, with an average of 83.82% (Table 2). All the RNA-Seq data in this study have been deposited in the NCBI SRA database (BioProject ID: PRJNA234461).

**Table 2.** Summary of transcriptome sequencing of Korean pine trees.

| Tree Age | Sample | Replicate | Raw Reads | Mapped Reads | Mapped Ratio (%) |
|---|---|---|---|---|---|
| 17 years | Pk-17B | 17B_1 | 2,684,675 | 2,028,943 | 75.57% |
| | | 17B_2 | 5,923,929 | 4,903,685 | 82.78% |
| | | 17B_3 | 11,475,000 | 9,610,718 | 83.75% |
| | Pk-17 | 17_1 | 4,801,139 | 3,994,016 | 83.19% |
| | | 17_2 | 26,660,412 | 22,411,979 | 84.06% |
| | | 17_3 | 28,346,897 | 24,913,370 | 87.89% |
| | Pk-17S | 17S_1 | 23,318,376 | 20,101,524 | 86.20% |
| | | 17S_2 | 29,793,382 | 25,787,332 | 86.55% |
| | | 17S_3 | 8,236,400 | 6,676,862 | 81.07% |
| 14 years | Pk-14B | 14B_1 | 5,632,200 | 3,975,935 | 70.59% |
| | | 14B_2 | 6,630,199 | 5,607,861 | 84.58% |
| | | 14B_3 | 23,648,879 | 20,657,509 | 87.35% |
| | Pk-14 | 14_1 | 15,976,748 | 13,601,374 | 85.13% |
| | | 14_2 | 14,994,418 | 13,041,940 | 86.98% |
| | | 14_3 | 18,435,392 | 15,239,213 | 82.66% |
| | Pk-14S | 14S_1 | 23,982,848 | 20,095,783 | 83.79% |
| | | 14S_2 | 27,427,279 | 22,772,330 | 83.03% |
| | | 14S_3 | 12,097,898 | 9,533,985 | 78.81% |
| 10 years | Pk-10 | 10_1 | 24,675,846 | 20,123,359 | 81.55% |
| | | 10_2 | 24,392,808 | 20,816,266 | 85.34% |
| | | 10_3 | 5,053,322 | 4,072,424 | 80.59% |
| 7 years | Pk-7 | 7_1 | 16,395,567 | 13,583,909 | 82.85% |
| | | 7_2 | 17,894,857 | 14,818,730 | 82.81% |
| | | 7_3 | 4,343,598 | 3,026,102 | 69.67% |
| 5 years | Pk-5 | 5_1 | 19,437,573 | 15,410,572 | 79.28% |
| | | 5_2 | 5,953,431 | 4,723,835 | 79.35% |
| | | 5_3 | 25,792,764 | 22,254,425 | 86.28% |
| Total | | | 434,005,837 | 363,783,981 | 83.82% |

Pk: *Pinus koraiensis* Sieb. et Zucc. Small cones were formed in the current year (2021), and big cones were formed in the previous year (2020). Pk-17S and Pk-14S were shade-grown trees, while others were open-grown trees. Pk-17B and Pk-14B were trees with seed cones, while others were trees without seed cones.

**Table 3.** Statistics of Korean pine transcriptome assembly and annotation.

| Assembly | | Number of Transcripts | Number of Unigenes | |
|---|---|---|---|---|
| | Number | 213,276 | 111,786 | |
| | Maximum transcript length | 17,887 | 17,887 | |
| | Minimum transcript length | 175 | 201 | |
| | Average transcript length | 902 | 716 | |
| | Median transcript length | 507 | 367 | |

**Table 3.** *Cont.*

| Annotation | Database | Number of Transcripts | Ratio (%) | Number of Unigenes | Ratio (%) |
|---|---|---|---|---|---|
| | N90 length of transcript | 353 | 276 | | |
| | N50 length of transcript | 1540 | 1254 | | |
| | blastx_swissprot | 70,232 | 32.93% | 29,389 | 26.29% |
| | protein | 159,537 | 74.80% | 76,500 | 68.43% |
| | blastp_swissport | 61,120 | 38.31% | 26,422 | 34.54% |
| | Pfam | 22,375 | 14.02% | 11,962 | 15.64% |
| | SignalP | 7038 | 4.41% | 3684 | 4.82% |
| | eggNOG | 93,802 | 58.80% | 41,194 | 53.85% |
| | blast2GO | 64,804 | 40.62% | 27,633 | 36.12% |
| | Pfam2GO | 11,740 | 7.36% | 6149 | 8.04% |
| | Kegg | 45,013 | 21.11% | 18,300 | 16.37% |
| | Total | 124,533 | 58.39% | 53,537 | 47.89% |
| **GO Analysis** | | **Number of Transcripts** | **Ratio (%)** | **Number of Unigenes** | **Ratio (%)** |
| Assigned to GO terms | | 52,244 | 24.50% | 21,070 | 18.85% |
| Assigned to "biological process" | | 29,995 | 14.06% | 12,214 | 10.93% |
| Assigned to "molecular function" | | 26,977 | 12.65% | 11,270 | 10.08% |
| Assigned to "cellular component" | | 41,933 | 19.66% | 17,050 | 15.25% |
| GO terms | | 4103 | 100% | 3692 | 100% |
| GO terms in the "biological process" category | | 2050 | 49.96% | 1830 | 49.57% |
| GO terms in the "molecular function" category | | 1560 | 38.02% | 1409 | 38.16% |
| GO terms in the "cellular component" category | | 493 | 12.02% | 453 | 12.27% |

After annotation, 124,533 (58.39%) transcripts had significant matches and at least one hit in the databases (Table 3). GO annotation was then applied to the 124,533 annotated transcripts in terms of "biological process", "molecular function", and "cellular component". In total, 52,244 (24.5%) transcripts were assigned to 4103 GO terms, including 21,070 (18.85%) unigenes (Table 3).

*3.3. Analysis of DEUs*

Based on 36 pairwise comparisons, a total of 934 DEUs were identified, and 568 DEUs were obtained after removing duplicates. At least one DEU was identified in 21 of 36 pairwise comparisons. The greatest number of DEUs was identified in the pairwise comparison Pk-17S vs. Pk-5, i.e., in the comparison between 5-year-old open-grown trees and 17-year-old shade-grown trees, showing that these two samples involve more biological processes in which larger numbers of genes take part (Table 4). However, no DEUs were identified in 15 pairwise comparisons (Table 4).

Among the 568 DEUs identified, 394 were annotated in at least one database, with FPKM ≥ 5 in at least one sample (Supplementary Table S1), and they were analyzed further based on their GO annotation. Here, the biological processes associated with environmental responses and sexual reproduction were emphasized (Supplementary Table S2).

**Table 4.** Statistics of differentially expressed unigenes (DEUs) in Korean pine trees.

| | Comparison | Number of DEUs | | |
|---|---|---|---|---|
| | | Up | Down | Total |
| 1 | Pk-7 vs. Pk-5 | 0 | 0 | 0 |
| 2 | Pk-10 vs. Pk-5 | 11 | 1 | 12 |
| 3 | Pk-14S vs. Pk-5 | 58 | 93 | 151 |
| 4 | Pk-14 vs. Pk-5 | 26 | 0 | 26 |
| 5 | Pk-14B vs. Pk-5 | 9 | 0 | 9 |
| 6 | Pk-17S vs. Pk-5 | 123 | 233 | 356 |
| 7 | Pk-17 vs. Pk-5 | 17 | 1 | 18 |
| 8 | Pk-17B vs. Pk-5 | 25 | 1 | 26 |
| 9 | Pk-10 vs. Pk-7 | 1 | 1 | 2 |
| 10 | Pk-14S vs. Pk-7 | 1 | 0 | 1 |
| 11 | Pk-14 vs. Pk-7 | 28 | 10 | 38 |
| 12 | Pk-14B vs. Pk-7 | 0 | 0 | 0 |
| 13 | Pk-17S vs. Pk-7 | 63 | 71 | 134 |
| 14 | Pk-17 vs. Pk-7 | 0 | 0 | 0 |
| 15 | Pk-17B vs. Pk-7 | 0 | 0 | 0 |
| 16 | Pk-14S vs. Pk-10 | 13 | 38 | 51 |
| 17 | Pk-14 vs. Pk-10 | 5 | 15 | 20 |
| 18 | Pk-14B vs. Pk-10 | 2 | 0 | 2 |
| 19 | Pk-17S vs. Pk-10 | 8 | 47 | 55 |
| 20 | Pk-17 vs. Pk-10 | 3 | 13 | 16 |
| 21 | Pk-17B vs. Pk-10 | 3 | 4 | 7 |
| 22 | Pk-17S vs. Pk-14S | 0 | 0 | 0 |
| 23 | Pk-17 vs. Pk-14S | 1 | 0 | 1 |
| 24 | Pk-17B vs. Pk-14S | 2 | 2 | 4 |
| 25 | Pk-17S vs. Pk-14 | 0 | 0 | 0 |
| 26 | Pk-17 vs. Pk-14 | 0 | 0 | 0 |
| 27 | Pk-17B vs. Pk-14 | 0 | 0 | 0 |
| 28 | Pk-17S vs. Pk-14B | 0 | 0 | 0 |
| 29 | Pk-17 vs. Pk-14B | 0 | 0 | 0 |
| 30 | Pk-17B vs. Pk-14B | 0 | 0 | 0 |
| 31 | Pk-14S vs. Pk-14 | 0 | 0 | 0 |
| 32 | Pk-14S vs. Pk-14B | 0 | 0 | 0 |
| 33 | Pk-14 vs. Pk-14B | 0 | 0 | 0 |
| 34 | Pk-17S vs. Pk-17 | 0 | 0 | 0 |
| 35 | Pk-17S vs. Pk-17B | 4 | 0 | 4 |
| 36 | Pk-17 vs. Pk-17B | 0 | 1 | 1 |
| Total | | 403 | 531 | 934 |
| Unique | | | | 568 |

Pk: *Pinus koraiensis* Sieb. et Zucc. Small cones were formed in the current year (2021), and big cones were formed in the previous year (2020). Pk-17S and Pk-14S were shade-grown trees, while others were open-grown trees. Pk-17B and Pk-14B were trees with seed cones, while others were trees without seed cones.

### *3.4. The Expression Patterns of DEUs Related to Environmental Responses*

During tree aging, age-related responses to climate [16] occur, enabling trees' survival in a complicated environment. Here, we identified 45 DEUs associated with responses to environmental changes, including "response to stress" (26 DEUs) and "response to light

and temperature" (19 DEUs); GO terms such as "defense response", "defense response to fungus", "response to heat", and "response to light stimulus" were also annotated (Supplementary Table S2). These data indicate the occurrence of transcriptomic changes in the environmental responses during Korean pine aging.

### 3.4.1. Response to Stress

Among the 26 DEUs associated with "response to stress", 10 had age-related expression patterns in these nine samples, and their expression patterns displayed two distinct clades. In clade one, galactinol synthase 4-1, 4-2, and oxidative stress response protein showed decreased expression levels during Korean pine aging, and their expression levels were lower in samples Pk-14S and Pk-17S than those in samples Pk-14 and Pk-17. In clade two, the other seven DEUs showed increased expression levels during Korean pine aging, and their expression levels were higher in samples Pk-14S and Pk-17S than those in samples Pk-14 and Pk-17 (Figure 2). These results further indicate that age-related stress responses occur in Korean pine trees, and shading influences the expression of genes associated with the stress response.

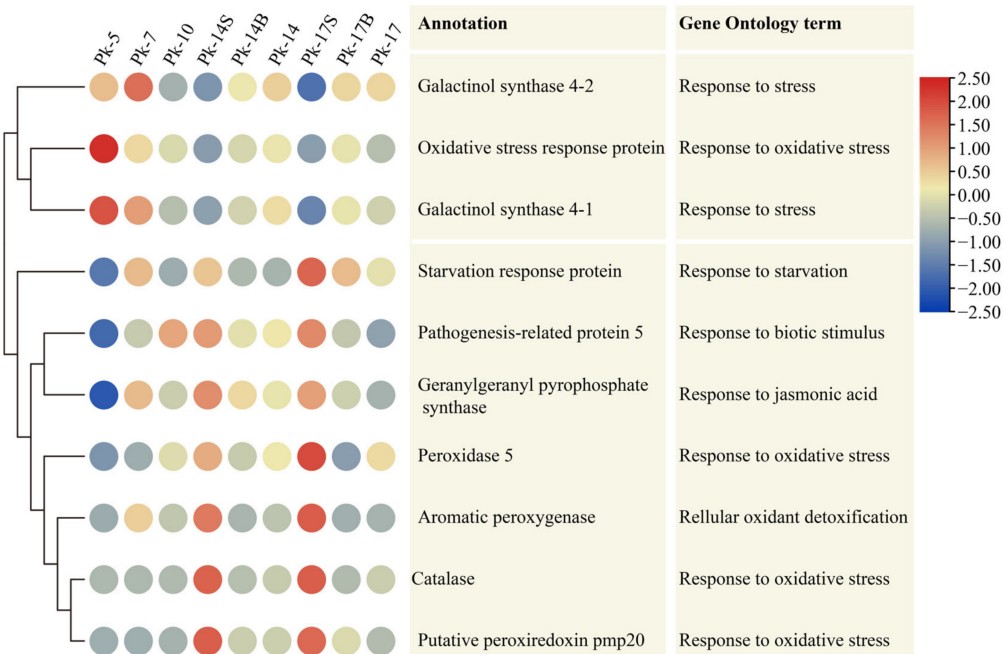

**Figure 2.** Heatmap showing the age-related expression patterns of 10 differentially expressed unigenes associated with the stress response, assayed by RNA-Seq, in Korean pine trees. The color scale (2.5 to −2.5) represents the values after they were log-scaled and row-scaled with fragments per kilobase of transcript per million mapped reads. Row clustering was performed according to the gene expression pattern.

### 3.4.2. Response to Temperature

Heat shock proteins function as molecular chaperones, and they are induced by heat stress [17]. Furthermore, they are activated by a range of stressors that are related to aging and the shortening of the life span, in addition to heat stress [18]. Here, we identified 15 heat shock protein genes from DEUs, and almost all of them showed the highest expression levels at age 5; their expression levels then decreased with age (Figure 3). To further confirm this result, we re-analyzed the larch transcriptomes generated in our previous work [19]. Notably, almost the same decreased expression patterns of heat shock protein genes with age were detected (Supplementary Table S3). These data further show that there may be a difference in the physiological response to temperature between juvenile and adult conifers. Moreover, the expression levels of these 15 Korean pine heat shock protein

genes were lower in samples Pk-14S and Pk-17S than those in samples Pk-14 and Pk-17 (Figure 3). These results indicate that shading influences temperature responses in Korean pine trees.

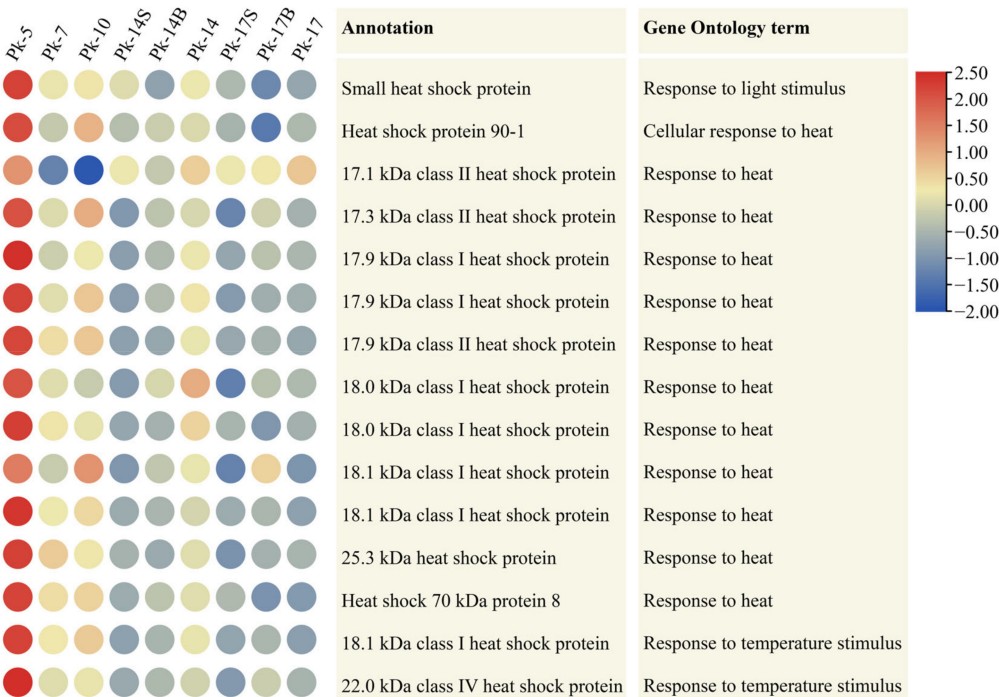

**Figure 3.** Heatmap showing the decreased expression patterns during tree aging of 15 differentially expressed unigenes annotated as heat shock proteins, assayed by RNA-Seq, in Korean pine trees. The color scale (2.5 to −2.5) represents the values after they were log-scaled and row-scaled with fragments per kilobase of transcript per million mapped reads.

### 3.5. The Expression Patterns of DEUs Related to Sexual Reproduction

Sensitivity to environmental signals that induce flowering increases with age [6,20,21]. To demonstrate the molecular aspects of this age-related physiological trait in Korean pine trees, we analyzed the expression patterns of DEUs associated with sexual reproduction. Among the 568 DEUs, 12 were assigned to GO terms associated with sexual reproduction, such as "sexual reproduction", "specification of floral organ identity", "pollen tube guidance", and "fruit ripening" (Supplementary Table S2), thereby showing the transcriptomic changes in sexual reproductive capacity during Korean pine aging.

One DEU (*PkDAL1*), annotated as *Pinus tabuliformis DEFICIENS-AGAMOUS-LIKE 1*, showed an age-dependent expression pattern during Korean pine aging (Figure 4). In samples Pk-5, Pk-7, Pk-10, Pk-14, and Pk-17, which were growing in open areas, its expression level increased with age (Figure 4a). In samples Pk-14 and Pk-14B, which had the same age and growth conditions but different reproductive states, its expression level was almost the same, and this was also found in samples Pk-17 and Pk-17B (Figure 4a). In samples Pk-14 and Pk-14S, which had the same age and reproductive state but different growth conditions, its expression level was almost the same, and this was also found in samples Pk-17 and Pk-17S (Figure 4a). In addition, there was a positive correlation between the expression pattern of *PkDAL1* and tree age ($R^2$ = 0.7604, Figure 4b).

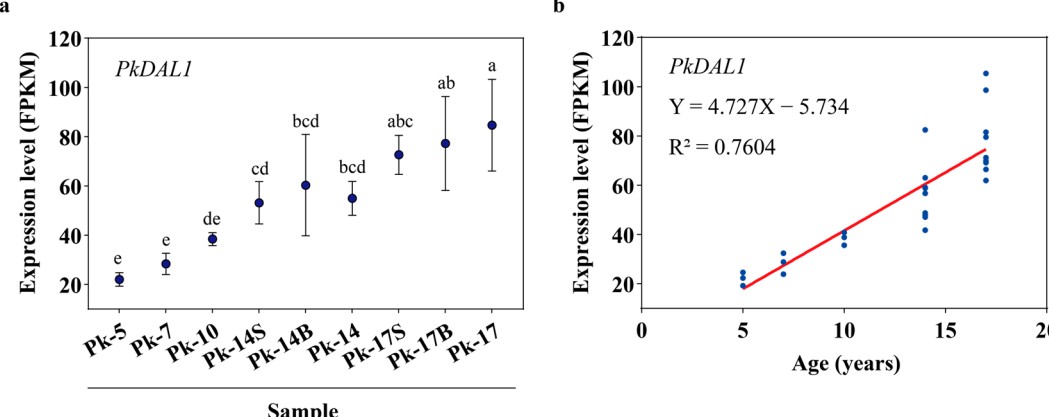

**Figure 4.** Plots showing the age-dependent expression pattern of *Pinus koraiensis DEFICIENS-AGAMOUS-LIKE 1* (*PkDAL1*), assayed by RNA-Seq. (**a**) *PkDAL1* expression levels shown as fragments per kilobase of transcript per million mapped reads (FPKM). Different lowercase letters above the plots indicate significant differences at the 0.05 level. Multiple comparisons after ANOVA, using the Duncan method, *n* = 3. Error bars, SD. (**b**) The relationships between *PkDAL1* expression patterns and age were calculated using the simple linear regression method.

## 4. Discussion

While this study focused on age and shading, other environmental factors, such as soil quality, water availability, and pest presence, may also significantly impact tree growth and physiology. In this work, the sampled trees grew closely, so other environmental factors had a lower impact on their growth and physiology. In the future, the influences of other environmental factors could be studied further. In conclusion, our findings offer valuable insights into how age and shading affect the development and physiological traits of Korean pine trees, providing a foundation for further studies and practical forest management strategies.

### 4.1. Transcriptomic Analysis Reveals the Age-Related Developmental and Physiological Changes in Korean Pine Trees at the Transcriptional Level

Traits like photosynthesis [22], the stress response [5], and reproduction [6] change dynamically during tree aging, and studying their underlying mechanisms helps us to understand tree adaptation and utilize this in forest management. In this study, based on transcriptomic analysis, DEUs associated with stress, light, temperature, and reproduction were identified from trees of different ages after pairwise comparisons. Our findings demonstrate that these traits change in Korean pine trees with age, it was possible to capture these changes via our transcriptomic analysis. More importantly, 10 DEUs associated with the stress response showed age-related expression patterns (Figure 2), indicating that stress responses might be mediated by these DEUs and may change as Korean pines age. Some DEUs are associated with the temperature response (Figure 3) and sexual reproduction (Figure 4), revealing that tree age coordinates these traits. Furthermore, studying the regulation of these DEUs will help us to understand how age controls these developmental and physiological traits.

### 4.2. Shading Affects the Expression of Physiological Response-Related Genes in Korean Pine

Light is a major environmental cue for plants [23], and a reduction in light intensity restrains plant growth in most species [24,25]. Plants respond to light in many ways and interact with the neighboring plants to avoid shading [12,26]. However, shading also protects seedlings from photo-oxidative damage in the early stage [27,28], and can even increase biomass and positively influence plant growth [29]. Altogether, the effect of shading on plant growth has two sides, and studying it is important for effective planting density adjustment, forest space allocation, and understory economic management [30].

Here, we provide molecular evidence that shading changes the physiological responses in Korean pine trees. For example, we identified 10 DEUs that were involved in the stress response, and their expression levels were changed by shading (Figure 2). Shading also decreased the expression levels of 15 heat shock protein genes (Figure 3). These results demonstrate that shading changes the expression of genes associated with stress, light, and temperature responses in Korean pine trees.

*4.3. The Expression Level of PkDAL1 Is Age-Dependent, Independent of the Reproductive State or Growth Conditions of Korean Pine Trees*

*PkDAL1* showed an increased expression pattern during Korean pine aging (Figure 4). In a previous work, a transcript (c77350.graph_c2) annotated as *Picea abies DAL1* was identified as being differentially expressed between 2- and 5-, 10-, 15-, or 30-year-old Korean pine trees [31]. The age-dependent expression patterns of *DAL1* homologs have also been characterized in *P. abies* [32], *P. tabuliformis* [33], and *Larix kaempferi* [34–36]. Here, we found that there was a positive correlation between the expression pattern of *PkDAL1* and tree age (Figure 4b), and its expression level was almost the same in trees of the same age but with different growth conditions or reproductive states (Figure 4a). Taken together, these results indicate that the expression level of *PkDAL1* is age dependent and independent of reproductive state or growth conditions. These findings are consistent with our previous study because there was no difference in *L. kaempferi DAL1* (*LaDAL1*) transcript levels between 13-year-old larch trees in different reproductive states [36], further confirming its role as an age marker in conifers.

Notably, overexpression of *DAL1* homologs promotes flowering in *Arabidopsis thaliana* [32,33,37,38]. In addition, *LaDAL1* accelerates the other life-cycle events in *Arabidopsis*, as well as flowering, by promoting the transition of meristem fate [38]. These studies indicate that the conserved function of *DAL1* homologs in the control of life-cycle progression may also exist in Korean pine trees.

**5. Conclusions**

By analyzing the differences in the transcriptomes of Korean pine trees of different ages, we can conclude that developmental and physiological traits like environmental responses and sexual reproduction change with tree age. We also found that shading changes the expression of genes associated with physiological responses in Korean pine trees. These findings not only reveal the effects of tree age and shading on tree development and physiology at the transcriptional level, but also provide a theoretical basis for Korean pine management.

**Supplementary Materials:** The following supporting information can be downloaded at: https://www.mdpi.com/article/10.3390/f15020291/s1. Supplementary Table S1. Summary of 394 differentially expressed unigenes with FPKM ≥ 5 in at least one sample. Supplementary Table S2. Expression patterns and annotation of 82 differentially expressed unigenes in Korean pine. Supplementary Table S3. Expression patterns of heat shock protein genes during larch tree aging.

**Author Contributions:** Z.-L.Y. conceived, designed, and carried out the study, analyzed the data, and wrote the manuscript. J.-Y.L. helped to carry out the study. J.F. provided suggestions on the experimental design and analyses. W.-F.L. conceived and designed the study, helped to analyze the data, and revised the manuscript. All authors have read and agreed to the published version of the manuscript.

**Funding:** This work was supported by the National Natural Science Foundation of China (32271904).

**Data Availability Statement:** All RNA-Seq data in this study have been deposited in the NCBI SRA database under accession number PRJNA234461.

**Acknowledgments:** The authors would like to thank Xiang-Yi Li (Research Institute of Forestry, Chinese Academy of Forestry) and the staff at Dabiangou Forest Farm for their help with sample collection. They also wish to thank I.C. Bruce (Peking University) for conducting a critical reading of the manuscript.

**Conflicts of Interest:** The authors declare no conflicts of interest.

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
