# Peer review of "Transcriptome Analysis Provides Insights into Korean Pine Tree Aging and Response to Shading"

_forests, doi:10.3390/f15020291_

Round 1

Reviewer 1 Report

Comments and Suggestions for Authors

This is an interesting research and the paper indicates scientific soundness. There are a few comments indicated in comment boxes in the attached file.

Reviewer 2 Report

Comments and Suggestions for Authors

This well-written and well-illustrated manuscript is of interest both on a practical level to determine tree physiological status and on a theoretical level to follow tree development. It would be interesting to follow the DEU markers in even older trees to see if they are even further expressed or suppressed. Is it possible that older trees without this pattern of expression are more susceptible to disease even before the trees are infected? Can this be used for forest management?

Reviewer 3 Report

Comments and Suggestions for Authors

In this work authors claim they performed RNA-sec of pine tree of different ages and lighting conditions. First of all the experimental design is false because it integrates a huge number of factors either environmental, genetical, soil condition, positioning and lighting. All these influence gene expression and the measured values will not be the consequence of only shading, but will include all other factors. And it would not be possible to isolate the contribution of lighting only. 

Secondly and most important: the data authors refer to is actually dated back to 2017, and cannot be collected “on the  morning of 31 July 2021”.  Moreover, samples (PRJNA234461) do not include those mentioned in the current paper. Therefore I rate the entire work as fake. 

The last author is also a co-author of a previously published paper based on that data (PRJNA234461). 

Following I will give some specific comments I have made before I found the falsification in the original data.

  • If “all trees were grown from seeds” Means there are big genetic diversity between the trees,  and how you will compare and isolate your lighting factor from other factors (here genetical). Additionally, growing conditions are actually very diverse between the trees 

  • “suggesting that at the age of 12, reproductive phase change had occurred in these three trees with cones, and it occurred in a genotype-dependent manner” - Your conclusion is based on nothing, nothing directs on genotype dependent changes. 

  • “mapping ratios ranged from 69.67% to 87.89% “ - This is actually very low probably because of different genetic background of trees 

  • Table 4: Pk-14S vs Pk-7 and Pk-14 vs Pk-7 - opposite results to 14 vs 5 years comparison. Actually this shows the data are random and chaotic.

Reviewer 4 Report

Comments and Suggestions for Authors

The study investigated the transcriptomic changes in Korean pine trees at various ages and under different growth conditions (open-grown and shade-grown), focusing on how age and shading impact tree growth and physiology. This concept is central to the "Afforestation under canopy" strategy used in regenerating Korean pine forests. The study used RNA-Seq to analyze the transcriptomic changes in the uppermost main stems of Korean pine trees aged 5, 7, 10, 14, and 17 years, both in open and shaded conditions.Key findings include the identification of 45 DEUs related to environmental responses, 15 heat shock protein genes indicating changes in temperature response with age, and 12 DEUs associated with sexual reproduction, highlighting age-related changes in reproductive capacity. Practical Implications: Findings from the study have practical applications in forestry, particularly in strategies like "Afforestation under canopy" which are crucial for the sustainable management of Korean pine forests.

The study is interest and the aims is clear. However, I have some concerns in this study. First, I am concerning the limited scope of age groups. The study focused on trees aged between 5 to 17 years. Including a broader age range, especially older trees, might have provided more comprehensive insights into age-related changes. More descript the life cycle of Korean pine trees may solve this point. Second, while the study focused on age and shading, other environmental factors like soil quality, water availability, and pest presence could also significantly impact tree growth and physiology. These factors were not explicitly considered. More discussion will be help in this point. In conclusion, this research offers valuable insights into how age and shading affect the development and physiological traits of Korean pine, providing a foundation for further studies and practical forest management strategies.

For the English writing, there are minimal errors, and the paper adheres to standard rules of English grammar, contributing to its professional quality. However, some sentences in the paper are quite long and complex. These sentences in the discussion and results sections are lengthy and complex, which might hinder readability. For instance, in the abstract, results and discussion sections, multiple ideas are often packed into one sentence. This can make the information challenging to follow. Breaking these into shorter, more concise sentences could improve clarity.

Comments on the Quality of English Language

For the English writing, there are minimal errors, and the paper adheres to standard rules of English grammar, contributing to its professional quality. However, some sentences in the paper are quite long and complex. These sentences in the discussion and results sections are lengthy and complex, which might hinder readability. For instance, in the abstract, results and discussion sections, multiple ideas are often packed into one sentence. This can make the information challenging to follow. Breaking these into shorter, more concise sentences could improve clarity.

Reviewer 5 Report

Comments and Suggestions for Authors

The authors investigated little-studied age-related changes in trees transcriptomes under different shading levels. The results obtained are original and interesting. However some MS details should be refined.

Firstly, please describe in more details procedures of total RNA, and mRNA purification, cDNA preparation and its amplification. How mRNA was fragmented, what was a size of fragments. As you know, details of samples processing may be crucial for interpretation of DEU results.

Lines 1328-29: “27 sets of sequencing reads were mapped to the assembled reference transcripts” change to “27 sets of sequencing reads were pairwise mapped to the assembled transcriptoms; in a result 36 comparisons were produced.”

Line 104: change libraryies to libraries

Correct the references list in accordance with the journal rules. (Author 1, A.B.; Author 2, C.D. Title of the article. Abbreviated Journal Name Year, Volume, page range.)

Round 2

Reviewer 3 Report

Comments and Suggestions for Authors

In the revised version and the cover letter authors simply repeat what was already in the text. The major issue is the experimental design allows a great level of uncertainty and randomness. Therefore, results are also random. My last comment shows that, on contradictory results from table 4. That was commented “results were based on the result of comparison”. Very funny. Nothing about the contradiction itself.

Authors claim that while the trees grow closely the environmental factors had lower impact on physiology. Table 1 shows a great difference, for example, in trees height in the same environment. Which means there are great difference between the trees both growing in shadow and on light. In such experimental settings, it is not possible to distinguish why such differences occurred. It can be genetical, can be environmental, now facts were presented in favor of genetics. The experimental design is false. I cannot see on the photo trees of different age, so they are growing far from each other, again fake?

In very simple case, you should show that you trees with cones have one genotype and trees without have another. Then you can say it is genotype-dependent. That would be a proof. Your speculations based on nothing.

It is very suspicious that a year is needed to release the data. I did that myself and it took a week. You claim in the paper the data is available, but in reality it is not. This is no way can be accepted. Please also include a count table together with raw reeds.

Reviewer 4 Report

Comments and Suggestions for Authors

The authors addressed on my suggestions and I don't have more question.